

# Remote sensing of land use/cover changes and its effect on wind erosion potential in southern Iran

Mahrooz Rezaei[1], Abdolmajid Sameni[1], Seyed Rashid Fallah Shamsi[2] and Harm Bartholomeus[3]

[1] Department of Soil Science, College of Agriculture, Shiraz University, Shiraz, Iran
[2] Department of Natural Resources Engineering and Environmental Sciences, College of Agriculture, Shiraz University, Shiraz, Iran
[3] Laboratory of Geo-Information Science and Remote Sensing, Wageningen University, Wageningen, The Netherlands

Corresponding author
Mahrooz Rezaei,
mahrooz.rezaei@gmail.com

## ABSTRACT

Wind erosion is a complex process influenced by different factors. Most of these factors are stable over time, but land use/cover and land management practices are changing gradually. Therefore, this research investigates the impact of changing land use/cover and land management on wind erosion potential in southern Iran. We used remote sensing data (Landsat ETM+ and Landsat 8 imagery of 2004 and 2013) for land use/cover mapping and employed the Iran Research Institute of Forest and Rangeland (IRIFR) method to estimate changes in wind erosion potential. For an optimal mapping, the performance of different classification algorithms and input layers was tested. The amount of changes in wind erosion and land use/cover were quantified using cross-tabulation between the two years. To discriminate land use/cover related to wind erosion, the best results were obtained by combining the original spectral bands with synthetic bands and using Maximum Likelihood classification algorithm (Kappa Coefficient of 0.8 and 0.9 for Landsat ETM+ and Landsat 8, respectively). The IRIFR modelling results indicate that the wind erosion potential has increased over the last decade. The areas with a very high sediment yield potential have increased, whereas the areas with a low, medium, and high sediment yield potential decreased. The area with a very low sediment yield potential have remained constant. When comparing the change in erosion potential with land use/cover change, it is evident that soil erosion potential has increased mostly in accordance with the increase of the area of agricultural practices. The conversion of rangeland to agricultural land was a major land-use change which lead to more agricultural practices and associated soil loss. Moreover, results indicate an increase in sandification in the study area which is also a clear evidence of increasing in soil erosion.

## INTRODUCTION

Wind erosion is a key problem in arid regions as a component of land degradation, which is not only closely related to geo-ecological factors but also to land use/cover changes and land management practices. Wind action in erosion, transport and subsequently deposition of fine particles, has been recognized as an important environmental problem

(*Goossens & Riksen*, *2004*). Two-thirds of Iran is located in an arid and semi-arid zone and more than half of the Iranian provinces are suffering from critical wind erosion (*Amiraslani & Dragovich*, *2011*; *Hui et al.*, *2015*).

Mensuration of wind erosion is not only important to understand wind erosion itself, but also an important scientific step in efforts to reverse the process of desertification (*Yue et al.*, *2015*). However, due to the complex inter-action of human–environment factors and wind erosion, it is difficult to be monitored and assessed. In such a context, erosion models can help to improve prediction and forecasting.

Over the past decades, several models have been developed to describe and estimate wind erosion potential, like the wind erosion equation (WEQ) (*Woodruff & Siddoway*, *1965*), Texas tech erosion analysis model (TEAM) (*Gregory et al.*, *2004*), and the wind erosion prediction system (WEPS) (*Hagen*, *1991*). These models need a variety of input data which limits their application in regions where this is sparsely available. Further, they are not optimized for the environmental and climatic conditions of Iran according to the employing factors required. In 1995, the Iranian Research Institute of Forests and Rangelands has developed an experimental model of wind erosion, named IRIFR (*Ahmadi*, *1998*). IRIFR considers the specific ecological conditions of this area, and can be used to estimate the potential wind erosion in central and southern Iran. The accuracy of the IRIFR model results has been assessed by field measurements using sediment traps (*Ahmadi*, *1998*).

Land use/cover change is one of the most sensitive indices of interactions between human activities and natural environment (*Minwer Alkharabsheh et al.*, *2013*). Therefore, in recent years, a number of studies have been carried out to estimate effects of land use/cover change on water erosion (*Martinez-Casasnovas & Sanchez-Bosch*, *2000*; *Szilassi et al.*, *2006*; *Cebecauer & Hofierka*, *2008*; *Garcia-Ruiz*, *2010*; *Wijitkosum*, *2012*; *Minwer Alkharabsheh et al.*, *2013*). All studies indicated a strong impact of land use/cover changes on water erosion and sediment transport rates. However, there are limited studies that investigate the influence of land use/cover changes on wind erosion.

Wind erosion is a key process in land degradation, but has not been studied well in relation with land use and associated land cover changes (*Li et al.*, *2014*). Soil physical and chemical characteristics, roughness, and land management practices are factors affecting erosion rates. Although the fundamental mechanism of wind erosion is the same for both rangelands and croplands (*Webb & Strong*, *2011*) these factors vary greatly between different land use/covers such as croplands and rangelands.

Facing vast areas of rapid changes, encouraged researchers to employ remote sensing techniques for spatially continuous and fast change detection of land use/cover. For decades, remote sensing has been extensively used for better understanding of land surface characteristics, dynamics and monitoring land use/cover changes (*Bartholome & Belward*, *2005*; *Gong et al.*, *2013*). Multispectral satellite data have proven to be a precious resource for monitoring land use/cover changes. Among the available multi-spectral imaging systems, the Landsat satellites have been widely used to derive information on land use/cover changes (*Gumma et al.*, *2011*; *Gong et al.*, *2013*; *Karnieli et al.*, *2014*).

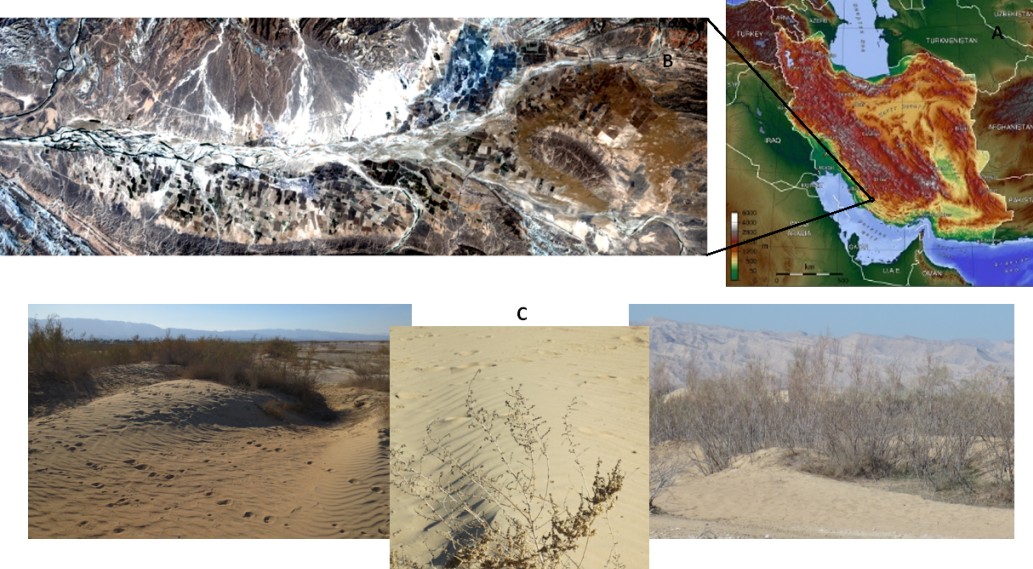

**Figure 1** Overview map (Image from Captain Blood/Wikimedia) (A) Image with the study area indicated and a true color composite of the 2013 Landsat 8 satellite image (B). Evidence of wind erosion in the study area (C).

The value of remote sensing data is enhanced through skilled interpretation, in conjunction with conventionally mapped information and ground-truthing (*Okin & Robert*, *2004*). However, due to the unpredictability of wind erosion events and often ephemeral nature of aftermath (*Clark et al.*, *2010*), it is generally difficult to assess wind erosion directly from remotely-sensed imageries. So, the main objective of this study is to assess the effect of land use/cover changes and land management practices on wind erosion potential during the previous decade in the southern Iran.

## MATERIALS AND METHODS

### Study area

The study area is located in the Fars province, in the southern part of Iran, (from 28° 07′15″ to 28° 13′07″N and 52° 07′36″ to 52° 23′55″E, covering an area of 17,260 ha), which is considered as the most critical wind erosion area of the province (Fig. 1). The study area is located in the Zagros geological zone, including Mishan, Aghajari, Bakhtiyari formations and Quaternary deposits. Soil of the study area is calcareous with Sandy loam and Loam texture.

The average altitude of the area is 211.5 m above sea level and the average slope is 0.84%. Mean annual maximum and minimum temperatures are 34 °C and 17 °C respectively, with an average of 25.5 °C. The area is facing a 190 mm average annual precipitation and 1,927 mm of average annual evapotranspiration (*Natural Resources and Watershed Management Office (NRWMO) of Fars province*, *2005*).

Soil temperature and moisture regime are hyper-thermic and aridic, respectively. The south western direction wind is the prevalent in the study area. The mean annual wind

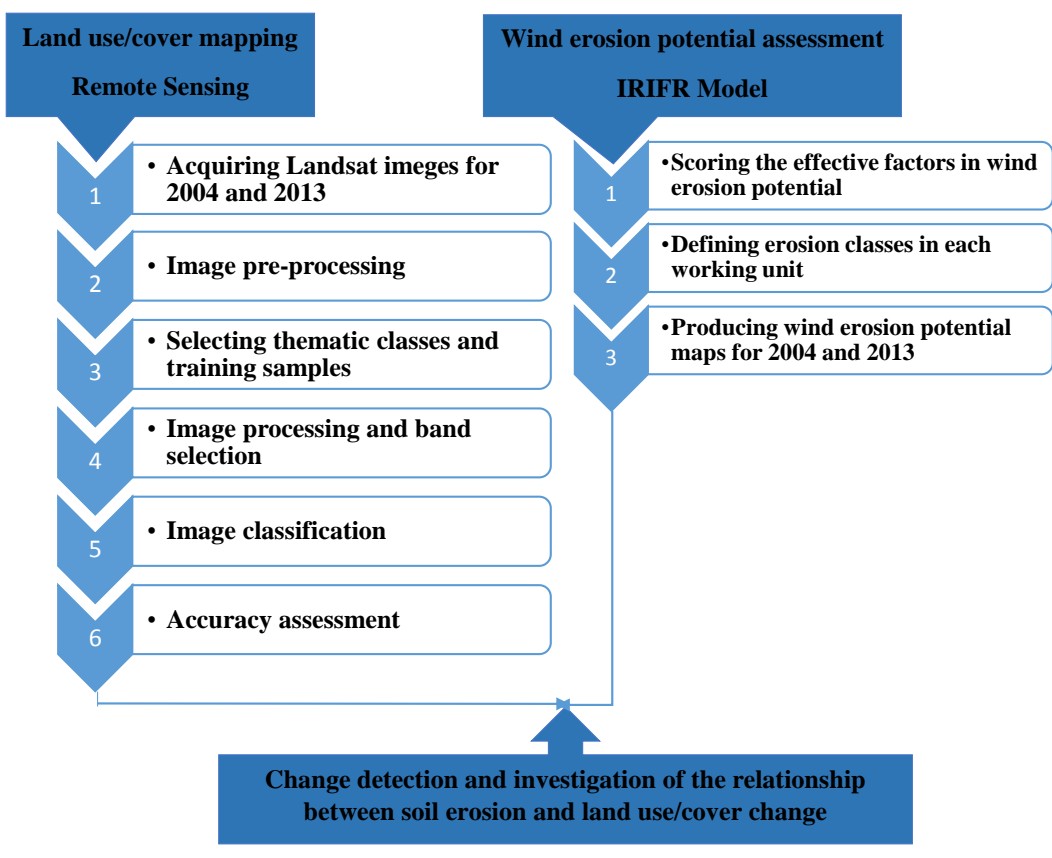

**Figure 2** Flowchart of the research.

speed at 2 and 10 m above the soil surface are 4.07 and 5.18 m s$^{-1}$ respectively. Wind erosion and dust storms are severe problems for the local inhabitants.

Rangelands and croplands are the most important land use/covers and land management practices in the region. The dominant plant species are *Salsola* sp., *Stipa capensis*, *Prosopis juliflora*, *Atriplex canescens* and *Haloxylon* sp. (*Natural Resources and Watershed Management Office (NRWMO) of Fars province*, *2005*).

## Assessing potential of wind erosion

The flowchart of the research is presented in Fig. 2. To choose a model to assess the potential wind erosion, data availability and costs have been taken into account. Therefore, the IRIFR model has been chosen due to its suitability for the ecological condition of Iran, data availability and running costs. IRIFR has two versions, developed for two types of landscape: IRIFR1 for non-arable and IRIFR2 for agricultural landscape.

Both IRIFR1 and IRIFR2, like the Pacific Southwest Interagency Committee (PSIAC) model for water erosion, are based on scoring 9 participating factors in the wind erosion process. These factors for IRIFR1 are: lithology, land form, wind velocity, soil and its surface cover, vegetation cover density, signs of soil surface erosion, soil moisture, soil type and distribution of aeolian deposits, land use and land management. The factors for IRIFR2 are: soil (sediment) texture, topography, wind velocity, soil roughness, crust and

**Table 1  Scoring the factors for IRIFR1.**

| No. | Factors | Range of scores |
| --- | --- | --- |
| 1 | Lithology | 0–10 |
| 2 | Land form (topography) | 0–10 |
| 3 | Wind Velocity | 0–20 |
| 4 | Soil surface cover | −5–15 |
| 5 | Vegetation cover density | −5–15 |
| 6 | Signs of soil surface erosion | 0–20 |
| 7 | Soil moisture | −5–10 |
| 8 | Type and distribution of wind deposits | 0–10 |
| 9 | Land use and land management | −5–15 |

**Table 2  Scoring the factors for IRIFR2.**

| No. | Factors | Range of scores |
| --- | --- | --- |
| 1 | Soil or sediment texture | 0–10 |
| 2 | Topography | 0–10 |
| 3 | Wind Velocity | 0–20 |
| 4 | Soil roughness | −5–15 |
| 5 | Crust and compressive stress of the soil | 0–20 |
| 6 | Soil moisture and irrigation status | −5–15 |
| 7 | Soluble salts in soil and irrigation water | 0–10 |
| 8 | Vegetation cover or residual density | −5–15 |
| 9 | Cropland management | −5–15 |

compressive stress of the soil, soil moisture and irrigation status, soluble salts in soil and irrigation water, vegetation cover or residual density, and cropland management.

These factors are scored according to their effects on sediment yield in a wind erosion process, shown for IRIFR1 and IRIFR2 in Tables 1 and 2, respectively (*Ahmadi*, *1998*). Range of scores in these tables guide the expert to give a score to each factor in each working unit. These ranges were introduced in IRIFR models according to several field studies in different parts of Iran and are based on experts' field knowledge, which was registered through questionnaires. The higher the score in each range the more potential for wind erosion. Moreover, a negative score in a factor indicates a negative effect on wind erosion. To produce a map of wind erosion potential based on the IRIFR model, it is necessary to define land units (LU), by overlaying geomorphological and land use/cover thematic maps as the first step. Then, in each LU, the factors were scored according to Tables 1 and 2.

The summation of the scores presents the wind erosion potential in the land unit. Finally, sedimentation yield is estimated using (Eq. 1), in which $Q_s$ is the total sediment yield in Tons $km^{-2} y^{-1}$ and $R$ is the summation of the 9 participating factors in the models.

$$Q_s = 41e^{0.05R}.$$ (1)

As shown in Tables 1 and 2, the most variable factor for both IRIFR1 & IRIFR2 is land use/cover and, consequently, land management practices. To predict any further change in potential of the study area for wind erosion, we considered change detection of land use/cover pattern and land management as the key factor of the models. To evaluate land use/cover a land management practices over a decade a variety of remote sensing techniques have been employed on Landsat-ETM+ and Landsat-8 images, of 2004 and 2013, respectively.

## Land use/cover mapping
### Image data
To assess the land use/cover Landsat L1T satellite images acquired on June 29, 2004 (Landsat Enhanced Thematic Mapper Plus (ETM+)) and acquired on June 30, 2013, (Landsat 8 Operational Land Imager (OLI) and Thermal Infrared Sensor (TIRS)), were downloaded from the USGS archives (http://earthexplorer.usgs.gov/).

### Image pre-processing
The downloaded images were geometrically corrected already, but pre-processing had to be done to ensure radiometric consistency between the images (*Koutsias & Pleniou*, *2015*). Therefore, a one-step radiometric correction using the dark-object subtraction method has been employed. This method is used to reduce the haze component in imagery caused by additive scattering from remote sensing data (*Chavez Jr*, *1988*). Using the dark-object subtraction method any value above zero in an area of known zero reflectance, such as deep water, represents an overall increase in values across the image and can be subtracted from all values in the corresponding spectral bands. Besides, regarding the SLC (Scan Line Corrector)-off problem of Landsat 7 images, Gap-Fill add-on in Envi software was used for filling the gaps.

Subsequently, histogram matching has been done between the two images. Digital values were extracted in the place of fifty random pixels over both image original bands before and after histogram matching. A paired sample $T$-test statistical analysis ($p < 0.05$) showed that the histogram matching was effective and has significantly changed the sampled digital values.

### Thematic classes and training sampling
According to the variations in land use/cover spectral behavior across the study area, it was difficult to define training samples representing thematic classes in a supervised classification procedure. Therefore, the selection of adequate and suitable training samples required an in-depth knowledge of the study area, which was achieved through an intensive field work and direct observation. A total of 127 points covering 12 land use/cover classes was collected using a handset global positioning system (GPS). The points were chosen in such a way that they adequately represented the variability of land use/cover spectral behavior in the study area. In addition, because of internal variability of certain thematic classes like agriculture, it was necessary to select some training samples for its subclasses. Residential areas were masked from the images.

Quality training samples were identified for the thematic classes, including rangeland, agricultural land with four subclasses, bare land, insusceptible areas with two subclasses, fan, residential area and others. Land use/covers related to wind erosion process such as Nebka and sand sheets were also included in the thematic classes, presenting wind erosion potential in the study area.

### Image processing and band selection

To investigate which combination of Landsat spectral bands yields the best classification results, we analyzed the performance of three different input band combinations for our classification: (a) the original spectral bands; (b) the first three principal components (PC-3); and (c) a combination of original and processed bands based on separability analysis.

For the last input data selection a variety of image processing and enhancement techniques was employed. The processed bands/indices included: band compositing, soil and vegetation indices calculation (VIs given in Table 4), principal component analysis (PCA), band fusion and texture analysis.

Next to calculation of VIs, texture analysis was employed using Variability, Fractal dimension, and Edge analysis methods, to detect areas that can be characterized by some form of repeating pattern on the ground. The edge analysis was done to provide convolution filters to enhance edge patterns in specific directions. Moreover, the prevalent wind direction can be taken into account via this analysis. Moreover, Gram–Schmidt spectral sharpening (*Laben & Brower*, *2000*), was performed to provide a higher resolution observation of the surface in a given period. These calculations were done using ENVI 5.1 and IDRISI taiga software.

To select the best combination of original and synthetic bands for final classification a hierarchical selection procedure was done, using the training dataset. At first, highly-correlated bands (correlation coefficient > 0.8) were removed to reduce repetitive information content. For example, the correlation coefficient of $SWIR_1$ and $SWIR_2$ was 0.96, thus only one of them ($SWIR_1$) was entered in the classification procedure. Further the main strategy of band selection was to select bands with a maximum separability for land use/cover classification. For this, statistical measures on the separability of signatures over a given set of bands were investigated. Further, the Digital Numbers of bands and calculated indices were plotted as a function of the band sequence producing a signature comparison chart (mean values) for the thematic classes using the SIGCOMP module in IDRISI. The bands in which the greatest separability among all classes occurred, were selected as optimal ones for recognition of these particular classes. For example among the soil and VIs, most of the classes were different from one another when looking at the WDVI, SI, YSI, LI indices, thus the other indices were removed from the classification procedure. Finally, the Transformed Divergence (TD) was calculated to assess the spectral separability of the training areas as shown in Table 6 (*Richards & Jia*, *2006*). In TD, we refer to 0 for the complete overlap between class pair signatures and to 2 for the total separation of the classes. The final training areas were selected by maximizing the separability metrics.

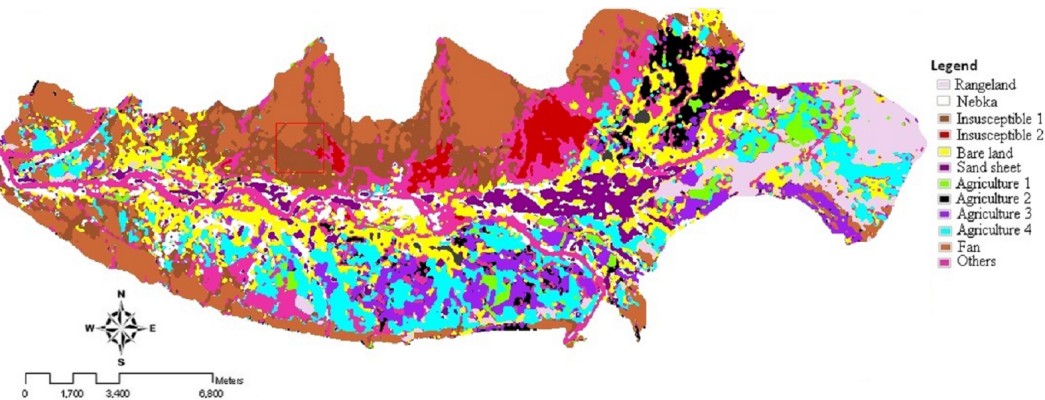

**Figure 3** **Land use/cover map of 2004, using ML rule.** Landsat imagery courtesy of NASA Goddard Space Flight Center and U.S. Geological Survey.

PCA was used to remove redundant information and was applied to solar-reflective spectral bands. The first three components described almost all the variance (Table 5).

### Image classification procedure

An integrated field survey and satellite remote sensing analysis was employed based on unsupervised and supervised image classification procedures (*Richards & Jia*, *2006*) to produce an accurate map of land use/cover changes and land management practices for the study area.

The classification scheme includes a preliminary analysis on both Landsat images. The same type of analysis for the two different Landsat scenes was carried out by testing the same combinations of classifiers with input data and training datasets. After the input bands were selected according to the method described in 'Image processing and band selection,' different supervised classification algorithms were tested, including Parallelepiped (PPD), Minimum Distance Classifier (MD), Mahalanobis Distance (MHD), and Maximum Likelihood Classifier (ML). It was found that Maximum Likelihood yielded the best results (see Table 7). Therefore, land use/cover map for 2004 and 2013 were produced using Maximum Likelihood (ML) rule of classification, shown in Figs. 3 and 4. In order to discriminate the river basin from bare land outside the river, river basin was masked and reclassified for further analysis. Thus, unclassified class in further tables are showing the bare land outside the river.

### Accuracy assessment

The Overall Accuracy (OA) and Cohen's Kappa coefficient (K), derived from the error matrix were used for the accuracy assessment of the final maps (*Congalton & Green*, *1999*). To generate a ground truth dataset, 94 locations were selected using random sampling, which were then visited to describe the land-cover type through field surveys. Next to this, the locations were controlled through visual interpretation of very high spatial resolution images that are available online on the Google Earth website.

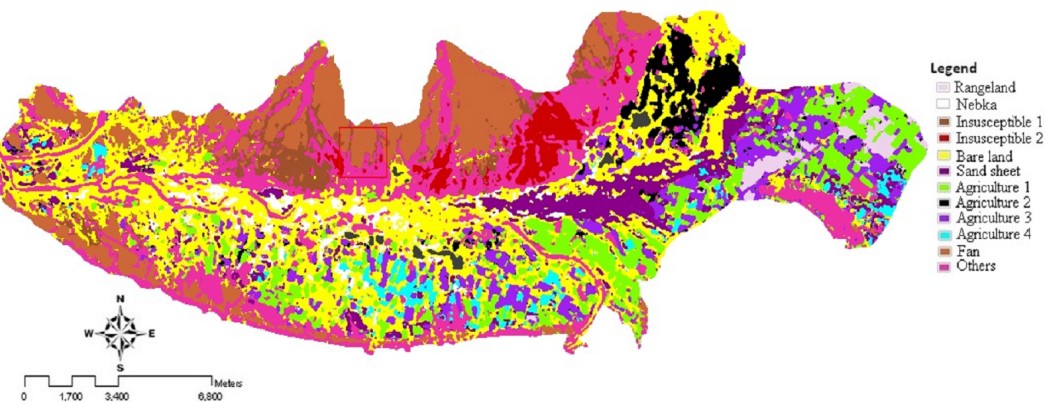

**Figure 4  Land use/cover map for 2013, using ML rule.** Landsat imagery courtesy of NASA Goddard Space Flight Center and U.S. Geological Survey.

## Change detection

Change analysis was performed by calculating cross-tabulation statistics derived from pair-comparison of classification results for 2004 and 2013. In addition, the analysis of the causes associated with the changes on soil losses was performed by cross-tabulation of the soil loss map and the map of land use/cover changes.

## RESULTS AND DISCUSSION

### Land use/cover changes from 2004 to 2013

The most accurate result was obtained using the selected combination of input data and ML classification algorithm (overall accuracy of 84% and 90.8% and Kappa coefficient of 0.8 and 0.9 for Landsat 7 and 8, respectively) for both Landsat images (Table 7). *Yousefi et al.* (*2015*) also found that ML algorithm is one of the best algorithms for land use mapping with average of 0.94 Kappa coefficient. According to the correlation and separability metrics, the near infrared (NIR) and short infrared band (SWIR1), the linear saturated thermal infrared band (TIR), the WDVI, SI, YSI, LI indices, and processed bands by edge analysis in the aspect of E-W and SE-NW were finally selected as the best input band combination. Land use/cover maps for 2004 and 2013 are shown in Figs. 3 and 4. Land use/cover variability of the study area comprised 12 classes. Table 8 indicates the area of each land use/cover class and its relative change during the period.

Several significant changes in land use/cover occurred between 2004 and 2013 (Table 9). These changes can affect soil loss due to wind erosion. The major reason is an increase of heavy and intensive grazing in rangelands, exposed to degradation for low-income agricultural activities and rain-fed farming. As shown in Table 9, rangeland is one of the most influenced land covers facing 76.19% of change. The results indicated that 55.22% of rangelands changed to agricultural lands. Moreover, 10.23% of these rangelands changed to sand areas in 2013. Low-efficiency irrigation systems combined with an increase in soil loss from arable lands leads to reduction in productivity. This is in line with findings by *Minwer Alkharabsheh et al.* (*2013*) who reported the progressive decrease of the agricultural areas and mixed rain-fed areas as the main reason of declining in soil erosion in Jordan.

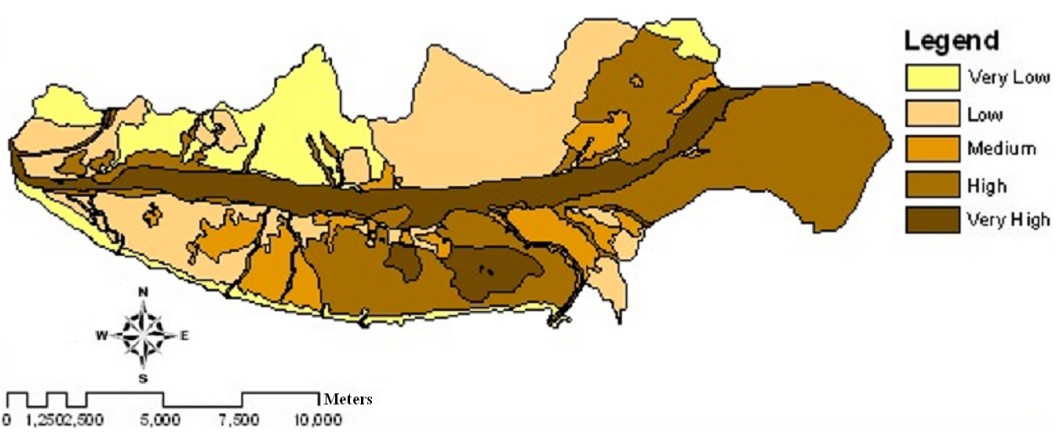

**Figure 5  Wind erosion potential map of the study area using IRIFR models in 2004.**

Sand sheets/bare sands were also facing a change of about 52.62% from 2004 to 2013, showing an expansion mostly to the southeastern parts. The sandification rate is an important index of land degradation, which involves aeolian erosion, windblown sands, shifting dunes and moving sands toward agricultural and residential areas (*Jiang*, *2002*; *Karnieli et al.*, *2014*). Nebkas decreased by 46.84 percent in the study area due to a decrease in vegetation cover. Moreover, the Nebkas were found to be unstable during field observations, therefore they have the potential to be blown away by wind and deposited at another location. Bare lands also increased significantly in 2013 compared to 2004. Bare lands or non-vegetation areas have a higher risk of soil erosion by wind than soil with a good vegetation cover. *Leh, Bajwa & Chaubey* (*2013*) also reported bare lands as one of the major source of increased erosion in the Ozark Highlands of the USA. In addition, residential areas increased by 91.64% in the study area between 2004 and 2013. In general, agricultural areas increased in the study area in 2013, and because of traditional cultivation methods in the study area, a short growing season which leads to short periods of soil surface cover, the absence of windbreaks, the wind erosion potential will be increased. Within the year, the difference in acquisition date of the satellite images which were used was just one day. Therefore, the changes in vegetation crown cover are probably not related to phenological differences within the growing season. During the long fallow stage, agricultural lands are without vegetation cover and farmers plough their fields several times during the rain events to increase the infiltration of the rainwater which eventually cause an increase in wind erosion potential.

In 2004, 2,079 $m^2$ of the study area abandoned from agricultural use, which decreased by 80 percent in 2013. The agricultural land-use change from abandoned land to arable land had an influence on wind erosion potential. In prior abandoned land, physical soil crusts developed more frequently. Usual mitigation measures by farmers has been tillage operations to reduce crusting, but this increased the potential of wind erosion (*Fister & Ries*, *2009*).

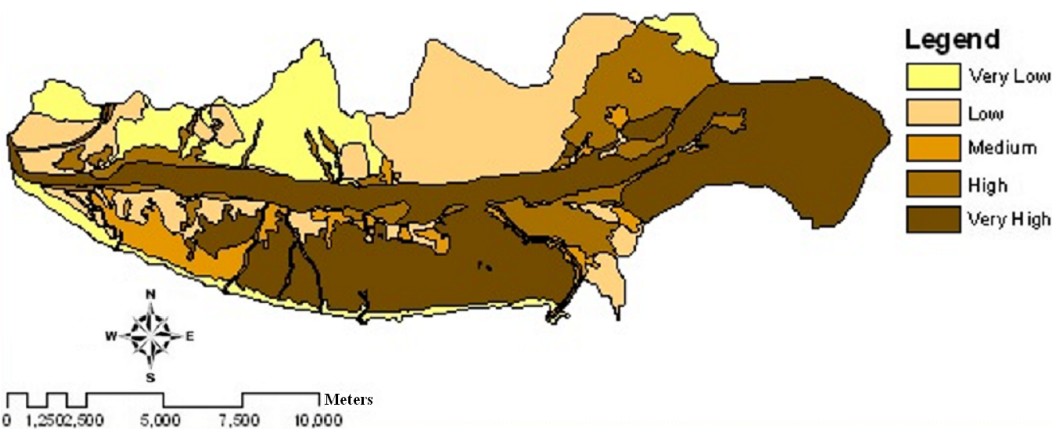

**Figure 6** Wind erosion potential map of the study area using IRIFR models in 2013.

**Table 3** Classes of wind erosion potential and estimated sedimentation potential for IRIFR1 and IRIFR2.

| Erosion class | Rate of erosion | Sum of scores | Sedimentation potential (Ton ha$^{-1}$ y$^{-1}$) |
| --- | --- | --- | --- |
| I | Very low | Less than 25 | Less than 2.5 |
| II | Low | 25–50 | 2.5–5 |
| III | Medium | 50–75 | 5–15 |
| IV | High | 75–100 | 15–60 |
| V | Very high | More than 100 | More than 60 |

## Changes in wind erosion potential from 2004 to 2013

Cross-tabulation between the 2 maps of wind erosion potential, shows the details of changes in each class (Table 10). Areas with very low and very high potential for wind erosion in 2004 did not show changes in 2013. On the other hand, 69.12 percent areas with high potential for wind erosion in 2004 changed to the very high sedimentation potential class in 2013. Moreover, 35.4 and 39.93 percent of the areas with medium wind erosion potential in 2004 changed into high and very high potential wind erosion classes in 2013, respectively. Furthermore, 11.69 and 0.76 percent of the areas with low sedimentation potential in 2004 changed to medium and high potential erosion classes in 2013, respectively. In general, results indicated that the wind erosion potential in the study area increased significantly in the period between 2004 and 2013.

The maps of wind erosion potential for the study area in 2004 (Fig. 5) and 2013 (Fig. 6) are compared in Fig. 8. Areas with the changes in wind erosion potential are shown in Fig. 7.

Table 3 and Figs. 5 and 6, indicate that the potential of sediment yield varies from 0.83 ton ha$^{-1}$ y$^{-1}$ to 272.71 ton ha$^{-1}$ y$^{-1}$ for 2004 and from 0.83 ton ha$^{-1}$ y$^{-1}$ to 350.16 ton ha$^{-1}$ y$^{-1}$ for 2013 in the study area. For very low and low levels of wind erosion potential, the potential sediment yield varies from 0.83 ton ha$^{-1}$ y$^{-1}$ to 4.52 ton ha$^{-1}$ y$^{-1}$. For the medium level wind erosion potential class, the potential sediment yield varies from 6.74 ton ha$^{-1}$ y$^{-1}$ to 15 ton ha$^{-1}$ y$^{-1}$. Moreover, for the high and very high level wind

**Table 4   Soil and Vegetation Indices (VIs).**

| No. | Index | Equation | Reference |
|---|---|---|---|
| 1 | Normalized difference vegetation index | $NDVI = (NIR - RED)/(NIR + RED)$ | *Rouse et al.* (*1974*) |
| 2 | Transformed vegetation index | $TVI = [(NIR - RED/NIR + RED) + 0.5]^{0.5}$ | *Deering et al.* (*1975*) |
| 3 | Corrected transformed vegetation index | $CTVI = [(NDVI + 0.5)/ABS*(NDVI + 0.5)]$ $.[ABS(NDVI + 0.5)]^{0.5}$ | *Perry & Lautenschlager* (*1984*) |
| 4 | Thiam's transformed vegetation index | $TTVI = [ABS(NDVI + 0.5)]^{0.5}$ | *Thiam* (*1997*) |
| 5 | Ratio vegetation index | $RVI = RED/NIR$ | *Richardson & Wiegand* (*1977*) |
| 6 | Normalized ratio vegetation index | $NRVI = (RVI - 1)/(RVI + 1)$ | *Baret & Guyot* (*1991*) |
| 7 | Soil adjusted vegetation index | $SAVI = (NIR - RED)/ (NIR + RED + L*).(1 + L)$ | *Huete* (*1988*) |
| 8 | Transformed soil adjusted vegetation index | $TSAVI = [a*(NIR - a.RED - b*)]/(RED + a.NIR - a.b)$ | *Baret, Guyot & Major* (*1989*) |
| 9 | Modified soil adjusted vegetation index | $MSAVI = [(NIR - RED)/(NIR + RED + L)].(1 + L)$ | *Qi et al.* (*1994*) |
| 10 | Weighted difference vegetation index | $WDVI = NIR - a.RED$ | *Richardson & Wiegand* (*1977*) |
| 11 | Difference vegetation index | $DVI = a.NIR - RED$ | *Richardson & Wiegand* (*1977*) |
| 12 | Perpendicular vegetation index | $PVI = [(RED_{soil} - RED_{veg})^2 + (NIRs_{oil} - NIR_{veg})^2]^{0.5}$ | *Richardson & Wiegand* (*1977*) |
| 13 | Normalized difference water index | $NDWI = (NIR - SWIR)/(NIR + SWIR)$ | *Cheng et al.* (*2008*) |
| 14 | Normalized difference salinity index | $NDSI = (RED - NIR)/(RED + NIR)$ | *Khan et al.* (*2001*) |
| 15 | Yazd salinity index | $YSI = (RED - BLUE)/(RED + BLUE)$ | *Dashtekian, Pakparvar & Abdollahi* (*2008*) |
| 16 | Salinity index | $SI = (SWIR1 - SWIR2)/(SWIR1 + SWIR2)$ | *Khaier* (*2003*) |
| 17 | Limestone index | $LI = (SWIR2^2 - NIR^2)/(SWIR2^2 + NIR^2)$ | *Mokhtari, Ghayumiyan & Feiznia* (*2005*) |
| 18 | Brightness index | $BI = (RED^2 + NIR^2)^{0.5}$ | *Khan et al.* (*2001*) |

**Table 5   Eigenvalues of the different eigen vectors after PCA for landsat 7 and 8, band 1 to 7.**

| Eigen vector | Variance (%) | |
|---|---|---|
| | Landsat 7 | Landsat 8 |
| 1 | 73.43 | 82.38 |
| 2 | 23.15 | 13.09 |
| 3 | 2.34 | 3.74 |
| 4 | 0.84 | 0.75 |
| 5 | 0.16 | 0.04 |
| 6 | 0.08 | 0.002 |
| 7 | – | 0.0003 |

erosion potential classes, the potential sediment yield varies from 21.29 ton ha$^{-1}$ y$^{-1}$ to 350.16 ton ha$^{-1}$ y$^{-1}$.

These results show that the area with a very high sediment yield potential increased, whereas the area with a low, medium, and high sediment yield potential decreased. The area with a very low sediment yield potential remained constant. 48.61% and 55.97% of the area include high and very high potential of wind erosion for 2004 and 2013, respectively.

Comparing land use/cover changes and corresponding wind erosion potential changes in 2004 and 2013 (Tables 11 and 12) indicated that soil wind erosion potential is

Rezaei et al. (2016), *PeerJ*, DOI 10.7717/peerj.1948

Table 6    **Transformed Divergence (TD) of the training set for Landsat7-ETM$^+$ and Landsat8- OLE imagery.**

| Training set | Rangeland | | Sand sheet | | Nebka | | Agi.1 | | Agri.2 | | Agri.3 | | Agri.4 | | Bare land | | Ins.1 | | Ins.2 | | Fan | | Others | |
|---|---|---|---|---|---|---|---|---|---|---|---|---|---|---|---|---|---|---|---|---|---|---|---|---|
| | L7 | L8 | L7 | L8 | L7 | L8 | L7 | L8 | L7 | L8 | L7 | L8 | L7 | L8 | L7 | L8 | L7 | L8 | L7 | L8 | L7 | L8 | L7 | L8 |
| Rangeland | | | 2 | 2 | 1.94 | 2 | 2 | 2 | 1.98 | 2 | 1.9 | 2 | 2 | 2 | 2 | 2 | 2 | 2 | 1.89 | 2 | 2 | 2 | 2 | 2 |
| Sand sheet | 2 | 2 | | | 2 | 1.98 | 2 | 2 | 2 | 2 | 1.99 | 2 | 1.98 | 1.9 | 1.96 | 1.99 | 2 | 2 | 2 | 2 | 2 | 2 | 2 | 2 |
| Nebka | 1.94 | 2 | 2 | 1.98 | | | 2 | 2 | 2 | 2 | 1.98 | 2 | 2 | 1.99 | 2 | 2 | 2 | 2 | 2 | 2 | 2 | 2 | 2 | 2 |
| Agri.1[a] | 2 | 2 | 2 | 2 | 2 | 2 | | | 2 | 2 | 2 | 2 | 1.99 | 2 | 2 | 2 | 2 | 2 | 2 | 2 | 2 | 2 | 1.99 | 2 |
| Agi.2 | 1.98 | 2 | 2 | 2 | 2 | 2 | 2 | 2 | | | 1.93 | 1.88 | 2 | 2 | 2 | 2 | 2 | 2 | 1.97 | 2 | 2 | 2 | 2 | 2 |
| Agri.3 | 1.9 | 2 | 1.99 | 2 | 1.98 | 2 | 2 | 2 | 1.93 | 1.88 | | | 2 | 1.89 | 1.82 | 1.79 | 2 | 2 | 2 | 2 | 2 | 2 | 1.96 | 1.96 |
| Agri.4 | 2 | 2 | 1.98 | 1.9 | 2 | 1.99 | 1.99 | 2 | 2 | 2 | 2 | 1.89 | | | 1.96 | 1.97 | 2 | 2 | 2 | 2 | 2 | 2 | 2 | 2 |
| Bare land | 2 | 2 | 1.96 | 1.99 | 2 | 2 | 2 | 2 | 2 | 2 | 1.82 | 1.79 | 1.96 | 1.97 | | | 2 | 2 | 2 | 2 | 2 | 2 | 1.99 | 2 |
| Ins.1[b] | 2 | 2 | 2 | 2 | 2 | 2 | 2 | 2 | 2 | 2 | 2 | 2 | 2 | 2 | 2 | 2 | | | 2 | 2 | | 2 | 1.98 | 1.99 |
| Ins.2 | 1.89 | 2 | 2 | 2 | 2 | 2 | 2 | 2 | 1.97 | 2 | 2 | 2 | 2 | 2 | 2 | 2 | 2 | 2 | | | 2 | 2 | 2 | 1.96 |
| Fan | 2 | 2 | 2 | 2 | 2 | 2 | 2 | 2 | 2 | 2 | 2 | 2 | 2 | 2 | 2 | 2 | 2 | 2 | 2 | 2 | | | 2 | 2 |
| Others | 2 | 2 | 2 | 2 | 2 | 2 | 1.99 | 2 | 2 | 2 | 1.96 | 1.96 | 2 | 2 | 1.99 | 2 | 1.98 | 1.99 | 2 | 1.96 | 1.8 | 1.79 | | |

**Notes.**
[a] Agri.1, 2, 3, and 4: Difference is based on land management.
[b] Ins.1, 2: Difference is based on the type of soil surface.

**Table 7** Overall accuracy and Kappa coefficient for the results of PPD, MD, MHD, and ML classification algorithms.

| Image | Algorithm | Overall accuracy | | | | Kappa coefficient | | | |
|---|---|---|---|---|---|---|---|---|---|
| | | PPD | MD | MHD | ML | PPD | MD | MHD | ML |
| **Landsat 7** | Spectral bands[a] | 50 | 56 | 58.4 | 78.3 | 0.43 | 0.47 | 0.47 | 0.67 |
| | PC-3 | 48.2 | 54.3 | 55.4 | 60 | 0.4 | 0.5 | 0.41 | 0.54 |
| | Selected inputs[b] | 75.6 | 56.5 | 76 | 84 | 0.66 | 0.5 | 0.65 | 0.8 |
| **Landsat 8** | Spectral bands | 57.4 | 71.4 | 78.3 | 80.1 | 0.53 | 0.67 | 0.74 | 0.74 |
| | PC-3 | 40 | 71 | 71 | 78 | 0.37 | 0.68 | 0.68 | 0.7 |
| | Selected inputs | 65.2 | 78.6 | 80 | 90.8 | 0.62 | 0.71 | 0.75 | 0.9 |

**Notes.**
[a] Spectral bands: Original bands of landsat 7 and landsat 8.
[b] Selected bands: Input band combination selected based on separability metrics.

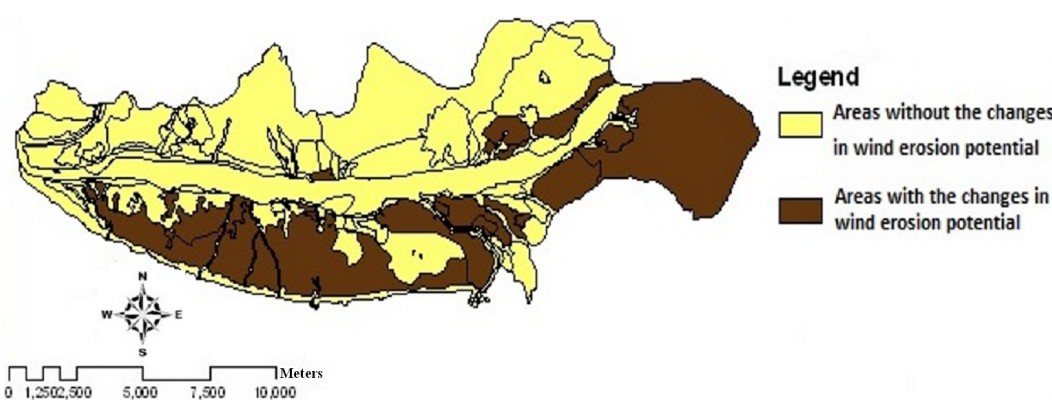

**Figure 7** The change of wind erosion potential between 2004 and 2013.

mainly increasing due to the changes in land use/cover in this period, since the other factors remained constant. Many researchers found that land use/cover change affects soil erosion positively and negatively. *Wijitkosum* (*2012*), studied the impact of land use/cover change on soil erosion in Pa Deng Sub-district, Thailand. He found that soil erosion decreased when land use/cover changed from bare land in 1990 to forest in 2010. *Yang et al.* (*2003*) indicated that with development of cropland in the last century, global soil erosion potential is estimated to have increased by about 17%. Moreover, *Sharma, Tiwari & Bhadoria* (*2011*) showed that transition of other land use/cover to cropland was the most detrimental to watershed in terms of soil loss.

Due to the low rainfall and high evapotranspiration, the study area has low vegetation cover and is susceptible to wind erosion even without human activities. However, according to the results obtained and field observations human activities including intensive livestock grazing, increasing cultivation, land-use change from rangelands to agricultural lands, and from abandoned land to arable land and using the underground water supply resulted in increasing soil loss due to wind erosion.

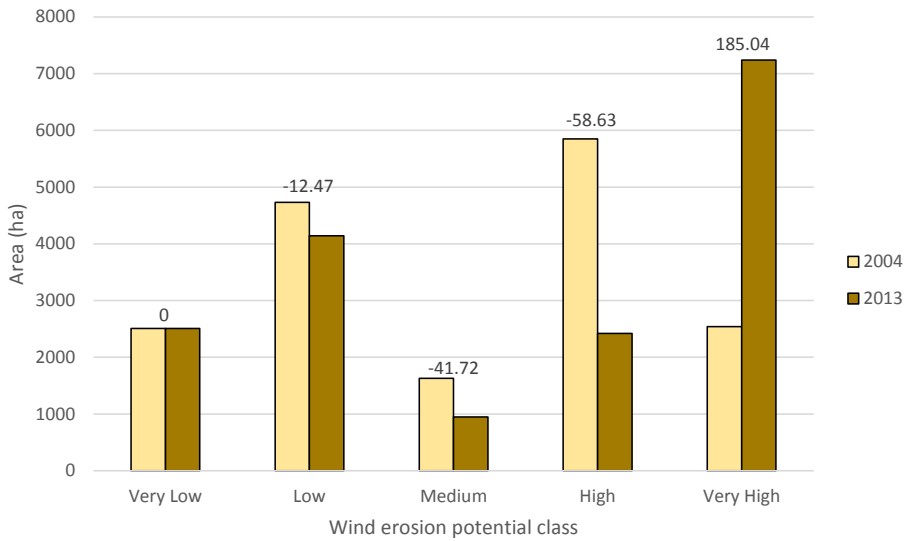

**Figure 8** Relative change in the area of wind erosion classes in 2004 and 2013.

**Table 8** Land use/cover of the study area in 2004 and 2013.

|  | Land use/cover | Area (ha) | | Relative change of land use/cover (%) |
| --- | --- | --- | --- | --- |
|  |  | 2004 | 2013 |  |
|  | Rangeland | 1,128 | 280 | −75 |
|  | Sand sheet | 854 | 1,303 | 52 |
|  | Nebka | 949 | 504 | −46 |
|  | 1 | 609 | 1,973 | 223 |
| **Agricultural land**[a] | 2 | 671 | 797 | 18 |
|  | 3 | 1,019 | 1,244 | 22 |
|  | 4 | 2,078 | 409 | −80 |
|  | Bare land (river basin) | 349 | 958 | 174 |
| **Insusceptible areas**[b] | 1 | 1,383 | 430 | −68 |
|  | 2 | 404 | 710 | 75 |
|  | Alluvial fan | 2,946 | 1,891 | −35 |
|  | Residential area | 50 | 96 | 91 |
|  | Others | 3,372 | 3,539 | 4 |
|  | Unclassified | 1,483 | 3,142 | 111 |

**Notes.**
[a] Agricultural lands: 1, High crop density; 2, Medium crop density; 3, Low crop density; 4, Abandoned lands.
[b] Insusceptible areas: 1, Calcareous Rocks; 2, Crusted areas.

Soil loss due to wind erosion from each land use/cover varies based on its characteristics like the vegetation cover type, surface roughness and management practices. The areas with an increase of soil erosion potential are located in the southern and northeastern parts of the study area. These areas mostly mainly exists of agricultural and rangeland. In these parts, extension of agricultural lands is obvious. The northern and northwestern

Rezaei et al. (2016), *PeerJ*, DOI 10.7717/peerj.1948

**Table 9   Matrix of changes in land use/cover (%).**

| | | 2004 | | | | | | | | | | | | | | Class total |
|---|---|---|---|---|---|---|---|---|---|---|---|---|---|---|---|---|
| | | Rangeland | Sand sheet | Nebka | Agri.1 | Agri.2 | Agri.3 | Agri.4 | Bare land | Ins.1 | Ins.2 | Fan | Residential area | Others | Unclassified | |
| | Rangeland | 23.81 | 0 | 0 | 4.03 | 0 | 0.12 | 0.42 | 0 | 0 | 0 | 0 | 0 | 0.02 | 0.1 | 100 |
| | Sand sheet | 10.23 | 47.84 | 16.1 | 4.03 | 1.29 | 4.2 | 9.19 | 19.79 | 0.6 | 0 | 0.47 | 0 | 5.12 | 8.09 | 100 |
| | Nebka | 0.09 | 9.07 | 11.57 | 0.49 | 0.24 | 0.65 | 2.09 | 12.94 | 0.1 | 0.29 | 0.02 | 0 | 4.18 | 4.87 | 100 |
| | Agri.1 | 36.92 | 5.71 | 0.62 | 25 | 6.88 | 29.5 | 23 | 1.65 | 1.36 | 0.09 | 1.41 | 0 | 6.8 | 10.05 | 100 |
| | Agri.2 | 1.75 | 0.35 | 0.38 | 12.5 | 46.6 | 3.67 | 2.81 | 0.13 | 0.04 | 0 | 2.29 | 0 | 2.13 | 11.48 | 100 |
| | Agri.3 | 16.53 | 2.61 | 3.33 | 28.6 | 8.9 | 24.5 | 16.8 | 1.03 | 0.28 | 0 | 0.54 | 0 | 1.58 | 6.48 | 100 |
| **2013** | Agri.4 | 0.02 | 0.87 | 0.94 | 2.61 | 2.59 | 11.4 | 6.58 | 0 | 0.3 | 0 | 0.6 | 0 | 0.72 | 2.07 | 100 |
| | Bare land | 0.04 | 14.45 | 26.2 | 1.82 | 2.88 | 0.34 | 0.04 | 58.48 | 0.03 | 0.27 | 0 | 0 | 8.82 | 0 | 100 |
| | Ins.1 | 0 | 0.01 | 0.09 | 0.16 | 0 | 0.01 | 0.06 | 0 | 44.8 | 0.09 | 4.69 | 0 | 1.02 | 0.02 | 100 |
| | Ins.2 | 0 | 0 | 0.24 | 0.13 | 0.03 | 0 | 0.28 | 0 | 6.56 | 87.9 | 0.49 | 0 | 7.1 | 0.04 | 100 |
| | Fan | 0.33 | 0.02 | 0.095 | 0.22 | 0.03 | 0.04 | 0.04 | 0 | 15.2 | 0.2 | 59.58 | 0 | 0.48 | 0.02 | 100 |
| | Residential area | 0 | 0.42 | 0.664 | 0.01 | 0.08 | 0 | 0.02 | 0 | 0 | 0 | 0 | 100 | 0.1 | 2.17 | 100 |
| | Others | 2.61 | 3.97 | 13.84 | 5.57 | 6.43 | 2.34 | 6.47 | 5.98 | 25.9 | 10.7 | 26.6 | 0 | 41.72 | 8.45 | 100 |
| | Unclassified | 7.67 | 14.68 | 25.93 | 14.9 | 24.1 | 23.2 | 32.1 | 0 | 4.94 | 0.42 | 3.31 | 0 | 20.21 | 46.16 | 100 |
| | Class total | 100 | 100 | 100 | 100 | 100 | 100 | 100 | 100 | 100 | 100 | 100 | 100 | 100 | 100 | |
| | Class changes | **76.19** | **52.16** | **88.43** | **75** | **53.4** | **75.5** | **93.4** | **41.52** | **55.2** | **12.1** | **40.42** | **0** | **58.28** | **53.84** | |
| | Image difference | **−75.13** | **52.62** | **−46.84** | **224** | **18.8** | **22.2** | **−80** | **174.59** | **−68.9** | **75.8** | **−35.8** | **91.64** | **4.93** | **111.86** | |

**Table 10** Changes between wind erosion potential classes in 2004 and 2013 (%).

| | | 2004 | | | | | Total |
|---|---|---|---|---|---|---|---|
| | | Very low | Low | Medium | High | Very high | |
| 2013 | Very low | 100 | 0 | 0 | 0 | 0 | 100 |
| | Low | 0 | 87.55 | 0 | 0 | 0 | 100 |
| | Medium | 0 | 11.69 | 24.67 | 0 | 0 | 100 |
| | High | 0 | 0.76 | 35.40 | 30.88 | 0 | 100 |
| | Very high | 0 | 0 | 39.93 | 69.12 | 100 | 100 |
| | Total | 100 | 100 | 100 | 100 | 100 | 100 |

parts showed no major change in sedimentation potential between 2004 and 2013. These parts includes areas that are not susceptible to wind erosion due the surface stoniness, hard soil surfaces (crust), which are not under cultivation. The central part (river basin), which is most of the time dry, shows a very high potential for wind erosion. The dry sediments inside the river basin are highly susceptible to wind erosion. In addition, lots of active Nebkas and sand sheets were found in the river basin that are ready to be transported by erosive wind.

The results indicated that the rangelands are susceptible to wind erosion. In total the area of rangeland decreased, and the potential for wind erosion in more than 90% rangelands increased to more than 60 ton ha$^{-1}$ y$^{-1}$ (from high to very high) in 2013, whereas, agricultural lands increased in 2013 and they showed high (15–60 ton ha$^{-1}$ y$^{-1}$) and very high (>60 ton ha$^{-1}$ y$^{-1}$) sedimentation potential. Sand sheets increased in 2013 in comparison to 2004 and their potential for wind erosion is mostly very high in both years. 94.97% of the sand sheets showed high and very high potential for wind erosion in 2013. For Nebkas, the sedimentation potential is also mostly very high and they showed a decrease in area in 2013. 63.7% and 72.41% of the Nebkas have a sedimentation potential of more than 60 ton ha$^{-1}$ y$^{-1}$ in 2004 and 2013, respectively.

River tributary (and bare land) showed an increase in 2013 and its potential for being eroded by wind was very high in both years. 96.52% and 87.02% of the bare lands showed a sedimentation potential of more than 60 ton ha$^{-1}$ y$^{-1}$ in 2004 and 2013, respectively. In general, seasonal rivers deposit their sediments in moist seasons with the sediments being transported by wind in dry seasons. These river beds are the most important eroding areas in Iran and most sand dunes have their source in river sediments in Iran (*Ahmadi*, *1998*).

More than 90 percent of the insusceptible areas and alluvial fans were found to have a very low and low sedimentation potential, due to their surface characterization discussed before.

It is obvious that with human activity like increasing agricultural lands, converting rangelands to cultivated lands, intensive grazing, and paying no attention to stabilizing sand dunes and Nebkas and climate factors including low rainfall, dry soil, and low vegetation coverage in the study area, soil wind erosion increased in the period between 2004 and 2013. These results indicate that policy and economic forces that shape land use

Rezaei et al. (2016), *PeerJ*, DOI 10.7717/peerj.1948

**Table 11 Cross-tabulation between land use/cover (in pixels numbers) and sedimentation potential in 2004.**

| Sedimentation potential (Ton ha$^{-1}$y$^{-1}$) | Land use/cover in 2004 | | | | | | | | | | | | | Total |
|---|---|---|---|---|---|---|---|---|---|---|---|---|---|---|
| | Rangeland | Sand sheets | Nebka | Agri.1 | Agri.2 | Agri.3 | Agri.4 | Bare land | Ins.1 | Ins.2 | Fan | Others | Unclassified | |
| <2.5 | 135 | 189 | 570 | 230 | 402 | 214 | 473 | 1 | 5,892 | 243 | 13,699 | 4,740 | 991 | 27,779 |
| 2.5–5 | 649 | 968 | 893 | 241 | 371 | 413 | 2,587 | 35 | 7,928 | 4,005 | 16,909 | 13,138 | 4,418 | 52,555 |
| 5–15 | 356 | 841 | 1,443 | 311 | 305 | 1,227 | 4,259 | 20 | 272 | 22 | 344 | 5,779 | 2,933 | 18,112 |
| 15–60 | 11,305 | 1,447 | 1,843 | 4,995 | 5,409 | 6,786 | 14,522 | 79 | 1,164 | 182 | 1,754 | 7,756 | 7,758 | 65,000 |
| >60 | 95 | 6,045 | 5,799 | 990 | 973 | 2,684 | 1,258 | 3,745 | 118 | 39 | 32 | 6,064 | 381 | 28,223 |
| Total | 12,540 | 9,490 | 10,548 | 6,767 | 7,460 | 11,324 | 23,099 | 3,880 | 15,374 | 4,491 | 32,738 | 37,477 | 16,481 | 191,669 |

Rezaei et al. (2016), *PeerJ*, DOI 10.7717/peerj.1948

**Table 12  Cross-tabulation between land use/cover (in pixels numbers) and sedimentation potential in 2013.**

| Sedimentation potential (Ton ha$^{-1}$y$^{-1}$) | Land use/cover in 2013 | | | | | | | | | | | | | Total |
|---|---|---|---|---|---|---|---|---|---|---|---|---|---|---|
| | Rangeland | Sand sheets | Nebka | Agri.1 | Agri.2 | Agri.3 | Agri.4 | Bare land | Ins.1 | Ins.2 | Fan | Others | Unclassified | |
| <2.5 | 0 | 148 | 107 | 310 | 36 | 139 | 11 | 16 | 2,782 | 621 | 9,028 | 12,239 | 2,342 | 27,779 |
| 2.5–5 | 0 | 114 | 233 | 441 | 130 | 805 | 416 | 49 | 1,643 | 6,592 | 11,215 | 14,909 | 9,453 | 46,000 |
| 5–15 | 0 | 466 | 491 | 600 | 69 | 424 | 66 | 35 | 104 | 2 | 373 | 3,392 | 3,840 | 9,862 |
| 15–60 | 1 | 2,994 | 716 | 3,837 | 6,802 | 1,100 | 209 | 1,283 | 246 | 677 | 163 | 3,457 | 5,403 | 26,888 |
| >60 | 3,118 | 10,762 | 4,060 | 16,734 | 1,828 | 11,364 | 3,852 | 9,271 | 9 | 2 | 237 | 5,329 | 13,878 | 80,444 |
| Total | 3,119 | 14,484 | 5,607 | 21,922 | 8,865 | 13,832 | 4,554 | 10,654 | 4,784 | 7,894 | 21,016 | 39,326 | 34,916 | 190,973 |

decision making can have impact on wind erosion and, importantly, emission of dust with local and regional consequences.

## CONCLUSION

Changes in land use/cover affect soil erosion considerably. These changes were especially increasing in agricultural lands and sandy areas. In order to reduce the potential of wind erosion, several practical works or guidelines can be considered:

- The seasonal river tributary which is one of the most important eroding areas in arid regions needs special attention from national and local governmental agencies for stabilizing shifting sands.
- Rangelands should be preserved from overgrazing and converting to low-income agricultural lands in order to decrease the unfavorable impact of cultivation practices on soil loss.
- To ensure a more efficient implementation of soil conservation in agricultural lands, a suitable agricultural practices must be applied. Perhaps tillage ought to be limited to periods with low wind velocities to minimize soil loss by wind erosion
- Cropping pattern and a crop calendar must be applied to decrease the long fallow stage and consequently to increase the vegetation coverage of the soil surface.

## ACKNOWLEDGEMENTS

The authors are grateful to the Natural Resources and Watershed Management Office of Fars province, Iran for providing thematic maps and technical advice.

### Funding

The authors received no funding for this work.

### Competing Interests

The authors declare there are no competing interests.

### Author Contributions

- Mahrooz Rezaei conceived and designed the experiments, performed the experiments, analyzed the data, contributed reagents/materials/analysis tools, wrote the paper, prepared figures and/or tables, reviewed drafts of the paper.
- Abdolmajid Sameni and Seyed Rashid Fallah Shamsi conceived and designed the experiments, performed the experiments, analyzed the data, contributed reagents/materials/analysis tools, wrote the paper, reviewed drafts of the paper.
- Harm Bartholomeus contributed reagents/materials/analysis tools, wrote the paper, reviewed drafts of the paper.

## Data Availability

Landsat data can be found at:

http://earthexplorer.usgs.gov/

Entity ID: LC81620412013181LGN00 for Landsat 8 (June 30 2013)

Entity ID: LE71620412004181ASN01 for Landsat ETM+ (June 29 2004)

Lat/long: from 28°07′15″ to 28°13′07″N and 52°07′36″ to 52°23′55″E, covering an area of 17,260 ha.

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
