# Peer review of "Remote sensing of land use/cover changes and its effect on wind erosion potential in southern Iran"

_PeerJ, doi:10.7717/peerj.1948_

## Round 0.1 · original submission · Major Revisions

The paper has been reviewed by 3 experts in the field. All agreed that it is an interesting study and worth to be published. Please address all of the referees' comment, especially Reviewer#1

Reviewer 1 ·

Basic reporting

The manuscript is mainly well written and the structure is clear, introduction and materials are well written, I added my comments into the text.
The chapter ""results" shoud be devided into 3 subchapters (I included my comments into the text). Figures should be added with respect to potential sediment yield etc. (see comments in the text).
A discussion is mssing and must be written taking the results of other authors into consideration.
The conclusions should be rewritten.

Experimental design

The experimental design is well planned and described, methods well decribed. Comments on details are included into the text.

Validity of the findings

The validity of the findings are well presented.

Additional comments

Please take the comments added into the manuscript into consideration and add a chapter discussion to the manuscript and rewrite the conclusions.

Reviewer 2 ·

Basic reporting

Line 75 & Lin 79: Citations of “Jia et al, 2015” and “Liu et al, 2015” are inappropriate. Globe land use/cover change database should be cited instead of a small region research.

Line 152, Fig.2: What’s the difference between Figure 2 and Figure 6 in line 374? If figures are the same, Figure 2 should be deleted.

Experimental design

The title of this manuscript is “Remote sensing of land use/cover changes and land management practices and its effect on wind erosion potential in Southern Iran”, but I can’t find any content about what and how land management practices effect wind erosion potential.

Line 312, Fig4: The legend (classification system of land use/cover) in Fig4 is very strange. What’s the meaning of “ Insusceptible 1” type ? What’s the difference between “ Insusceptible 1” and “ Insusceptible 2”? So do “Agriculture 1” to “4”. Alluvial fan is a type of geomorphology, not a type of land use/cover. A detailed description of those land use/cover types should be added.

Landsat 7 has been out of work for a long time, did you carry out any procedures to fill those gaps? And I think it’s meaningless to compare with Landsat 8.

Validity of the findings

Content of land use/cover classification is too much in the manuscript. Authors should focus on the effect on wind erosion potential by land use/cover changes and land management practices.

Additional comments

No Comments

Reviewer 3 ·

Basic reporting

The basic reporting is good.

Experimental design

No comments

Validity of the findings

The validity of the findings is fair.

Additional comments

The manuscript by Mahrooz and coauthors examines effects of land use land cover changes on wind erosion over Iran. Using high resolution remote sensing data authors show evidence for soil erosion due to increase in agriculture practices area in 2013 when compared to 2004. This paper systematically reports the observed evidence on increased soil erosion. The authors also show that the main contributor for increased soil erosion is conversion of range-land into agricultural practices. I find that the presentation is good and the results will be of interest to scientific community in the field. I recommend publication of the manuscript.

Minor comments:

1) Line 29: I would not say 'was an important land use change'. I suggest authors to phrase this differently. May be ' major land use change'?


2) In Table 1 and Table 2: What does 'Range of scores' represent. For example: Why Lithology score is 0-10, whereas land use and land management score is -5 to 15. I do not see any explanation on 'range of scores' and what does their magnitude represents?

---

## Round 0.2 · accepted · Accept

The paper has been revised accordingly. The 2 reviewers added minor comments that can be incorporated in the final publication.

Reviewer 2 ·

Basic reporting

no

Experimental design

no

Validity of the findings

no

Additional comments

Authors have revised this manuscript well based on comments, and I think manuscript can be accepted after a minor revision. Citations of Jia et al. 2015 and Liu et al. 2015 were deleted in Introduction, but still remain in References. They should be removed.

Reviewer 3 ·

Basic reporting

"No Comments".

Experimental design

"No Comments".

Validity of the findings

"No Comments".

Additional comments

Manuscript is much improved and I suggest authors to add also what does the negative score means in the manuscript around the lines 126-131.